# Titanium Dioxide Nanoparticles Doped with Iron for Water Treatment via Photocatalysis: A Review

**DOI:** 10.3390/nano14030293

**Published:** 2024-01-31

**Authors:** Domenico Rosa, Nigar Abbasova, Luca Di Palma

**Affiliations:** 1Department of Chemical Engineering Materials Environment, Sapienza-Università di Roma, Via Eudossiana 18, 00184 Roma, Italy; domenico.rosa@uniroma1.it; 2Department of Ecology, Azerbaijan University of Architecture and Construction, AZ1073 Baku, Azerbaijan; nigar.abbasova.bdu@gmail.com

**Keywords:** titanium dioxide, doping, iron, photocatalysis, nanoparticles

## Abstract

Iron-doped titanium dioxide nanoparticles are widely employed for photocatalytic applications under visible light due to their promising performance. Nevertheless, the manufacturing process, the role of Fe^3+^ ions within the crystal lattice of titanium dioxide, and their impact on operational parameters are still a subject of controversy. Based on these assumptions, the primary objective of this review is to delineate the role of iron, ascertain the optimal quantity, and elucidate its influence on the main photocatalysis parameters, including nanoparticle size, band gap, surface area, anatase–rutile transition, and point of zero charge. Moreover, an optimized synthesis method based on comprehensive data and insights from the existing literature is proposed, focusing exclusively on iron-doped titanium oxide while excluding other dopant variants.

## 1. Introduction

Heterogeneous photocatalysis, using semiconductor nanoparticles, is a promising technology for the removal of bio-refractory substances from contaminated water [1]. The development of nanostructured photocatalysts with enhanced and implementable structural, morphological, and electronic properties [2,3] proved to be effective in the treatment of industrial wastewater.

The efficacy of heterogeneous photocatalysis lies in the ability to harness sunlight or alternative light sources to activate catalytic processes capable of removing contaminants from water [4,5,6,7]. The semiconductors exhibit a valence band (VB) filled with electrons alongside a conduction band (CB) characterized by higher energy and an absence of electrons, establishing a well-defined band gap. Heterogeneous photocatalysis employed for water decontamination from organic substances involves the absorption of light with a specific wavelength by the photocatalyst. This process induces the excitation of electrons, facilitating their transition from the valence band to the conduction band of the photocatalyst. Consequently, this electron movement generates positive holes (h_VB_^+^) within the catalyst’s VB, possessing a potent oxidative potential. These positive holes effectively reduce adsorbed H_2_O molecules, producing free hydroxyl radicals OH, while electrons (e_CB_^−^) in the CB, notwithstanding their lower reduction potential compared to the oxidation potential of positive holes [8], are caught by oxygen molecules, leading to the formation of the superoxide radical anion O_2_^−^·. Free hydroxyl radicals stand out as powerful oxidizing agents, with an oxidation potential of 2.80 V: they can mineralize any organic compound adsorbed on the particle surface to yield water and carbon dioxide [4,5,9]. The direct oxidation effect of organic compounds by positive holes has also been hypothesized (any species with unpaired electrons or conjugated π bonds can be subject to oxidation by photogenerated positive holes) [4,10].

Surface OH groups actively take part in the photodegradation process by acting as trapping sites for h_VB_^+^, leading to the formation of highly reactive hydroxide radicals. Alternatively, these groups may facilitate the absorption of substrate molecules. Figure 1 provides a schematic representation of the photochemical activation of titanium dioxide and the consequent formation of hydroxyl radicals.

Despite considerable efforts devoted to the investigation of photocatalytic mechanisms, the practical implementation of photocatalysis at the industrial scale remains circumscribed. The main issue limiting the transition from laboratory experimentation to large-scale application is the lack of facile and environmentally sustainable methodologies for the large-scale production of high-performance photocatalysts under visible light [11].

## 2. Advanced Photocatalyst

Photocatalytic materials have gained considerable interest due to their numerous applications. Through appropriate modifications, these materials can effectively overcome limitations in process efficiency [12]—hence the term “advanced catalysts”.

Their great advantage is that they can harness solar radiation. For photocatalysts such as titanium dioxide (titania, TiO_2_), the most exhaustive scrutiny has been carried out [13,14,15] due to its cost-effectiveness [16], stability [17,18], widespread availability, and heightened activity [9,19]. A modification is essential to enhance its photochemical and adsorptive performances, especially to be active under visible light [13,20]. The primary drawback of titania is that its activity is only limited to UV radiation (λ ≤ 387 nm) due to its wide bandgap (3.2 eV) [21]. Only UV photons have sufficient energy to overcome this bandgap, thus promoting photocatalysis [20]. However, titania possesses a high rate of recombination of photogenerated electron-hole pairs, significantly limiting achievable quantum yield [22].

Consequently, doping titania nanoparticles with heteroatoms to reduce the bandgap, shifting the absorption edge towards longer wavelengths, has become a common practice [23]. This modification allows the photocatalyst to be active under visible light, minimizing charge recombination and achieving the ideal nanoscale size for photocatalytic applications [22]. It is widely accepted that a lower recombination rate strongly improves photocatalytic efficiency [23].

Among potential heteroatoms, iron is highly favored for doping titania [24,25], as shown in Figure 2, due to its cost-effectiveness [26] and efficacy. Additionally, its atomic radius, similar to titanium (0.61 Å for Ti^4+^ hexacoordinated and 0.55 Å for Fe^3+^ hexacoordinated) [27], facilitates the simple substitution of iron into the crystalline lattice of titanium [28,29]. While other authors may report slightly different values, they remain closely comparable, such as 0.645 Å for Fe^3+^ and 0.604 Å for Ti^4+^ [30,31] or 0.64 Å for Fe^3+^ and 0.68 Å for Ti^4+^ [32,33].

Figure 2 shows the number of publications related to various transition metals, including Fe, Cu, Zn, Cr, Ni, Co, and Mn, particularly in the framework of photocatalysis processes employing titanium dioxide. Furthermore, it also provides the total number of publications associated with iron-doped titania across all applications. Most of the publications on iron-doped titania are focused on its utilization in photocatalysis, constituting over half of all studies on iron-doped titania. Indeed, in 2023, there were 5716 publications on iron-doped titania, with 3757 exclusively dedicated to photocatalysis, representing 66% of all publications on iron-doped titania.

The extensive literature on iron-doped titania for photocatalysis purposes underscores its significance and appeal within the scientific community. This trend not only highlights the prevalence of iron-doped titania in photocatalytic applications but also shows its comparative advantage over alternative transition metal dopants, especially in the context of water treatment and environmental remediation.

Understanding the pivotal role of iron (Fe^3+^) in the improvement of titania photocatalytic properties requires examination of key photocatalysis steps: (a) the photogeneration of positive hole–electron pairs, necessitating trapped/separated to prevent recombination; (b) reduction and oxidation reactions involving distinct electrons and holes interacting with appropriately adsorbed species; and (c) the progression of intermediates, desorption of products, and surface reconstruction. Iron predominantly influences steps (a) and (b) in a complex manner, demanding comprehensive exploration and discussion for optimizing the photocatalytic process [34].

It is worth emphasizing that the concept of doping is sometimes interpreted literally as the integration of isolated heteroatoms (in this case, Fe^3+^) into a solid solution within the TiO_2_ framework. However, it is commonly used to describe the deposition of small oxide particles or, more generally, a secondary phase (e.g., Fe_2_O_3_) onto the surface of TiO_2_. Nonetheless, the mechanism for enhancing the photochemical response of TiO_2_ differs significantly between surface-deposited or doped Fe^3+^/TiO_2_ or Fe_2_O_3_/TiO_2_. In the former case, a reduction in the bandgap of the photocatalytic semiconductor occurs, while in the latter, a charge transfer to TiO_2_ from the deposited oxide is realized, known as coupling [35].

However, as discuss more comprehensively in this review, distinguishing between these mechanisms can sometimes be challenging. Nevertheless, it is crucial to do so to justify the occasionally conflicting properties of the materials; indeed, the catalytic role during photooxidation processes remains controversial due to the significant structural modifications it can induce in the photocatalyst [36]. These alterations concerning nanoparticle size, the anatase-to-rutile phase transition, bandgap, surface properties, etc., are often interrelated and can either enhance or diminish photocatalytic performance [37,38,39]. For instance, Fe^3+^ cations are thought to act as scavengers for photogenerated electrons during the photocatalytic process in the titania structure [40]. However, discrepancies emerge concerning their impact on enhancing electron-hole recombination properties [23]. Some authors propose that Fe^3+^ serves as a trapping site for both electrons and positive holes, especially at high concentrations [41]. Conversely, other studies show that Fe^3+^ exclusively traps positive holes, with electrons being captured by surface Ti^4+^ [42], or that it promotes charge separation [23].

Li X. et al. (2003) found that Fe^3+^ can trap photogenerated charges at different sites simultaneously. Positive hole trapping is specifically linked to surface Fe^3+^, whereas electron trapping is associated with bulk Fe^3+^ [43].

In general, achieving optimal photocatalytic properties typically involves doping at a relatively low level, usually below 1% [22]. However, the optimal dosage depends on nanoparticle size, diminishing as particle size increases. This trend arises because, in larger particles, charge recombination predominantly occurs in the bulk, whereas in smaller particles, it primarily takes place on the surface [42]. This phenomenon is linked to the recombination process dynamics, which can be associated with the distance between dopant cations in the titania structure [44]. Furthermore, changes in surface properties induced by doping can significantly contribute to explaining the photocatalytic performance of the system. The photocatalytic behavior of a material is determined by various physical, chemical, and electronic parameters that are often challenging to correlate. Photoactivity results from a balance of these factors, some of which frequently play conflicting roles [45].

To enhance photocatalytic performance, it is crucial to carry out a thorough investigation into how the quantity of iron affects the properties of the photocatalyst and, consequently, its photocatalytic activity. This relationship is intimately linked to morphological properties, as the photocatalytic activity is intricately correlated with these characteristics [29,46].

## 3. Modification of Titanium Dioxide by Iron

### 3.1. The Size of Nanoparticles

Titanium dioxide nanoparticles exhibit a spherical morphology [47,48], with their sizes dependent on the synthesis method employed. Common methods include sol-gel, co-precipitation, and hydrothermal approaches, with synthesis temperature also impacting nanoparticle sizes [29].

Nanoparticle size holds significant importance for catalysis, influencing the specific surface area of a catalyst. Smaller particle sizes enhance the number of specific active surface sites, thereby promoting charge carrier transfer in photocatalysis. However, Z. Zhang et al. (1998) noted that photocatalytic efficiency does not monotonically increase with decreasing particle size. An optimal size of approximately 10 nm was identified for pure nanocrystalline TiO_2_ photocatalyst in the liquid-phase decomposition of chloroform due to the antagonistic effect of increased surface recombination of the positive hole–electron pair [42]. C. Almquist (2002) observed a similar phenomenon, noting significant variation in photocatalytic performance below 30 nm due to reduced likelihood of charge recombination, with an optimal size of approximately 25 nm [49]. For particles exceeding this value, the available surface area for redox reactions diminishes as particle size increases. The introduction of iron, typically in the form of iron ions (e.g., Fe^3+^), reduces the size of titanium dioxide nanoparticles [50], as depicted in Figure 3.

For instance, I. Mwangi et al. (2021) reported an average diameter reduction from 13.7 nm for pure titania to 10.8 nm for iron-doped samples [47]. Similarly, Z. Ambrus et al. (2008) observed a decrease in nanoparticle size from 30 nm to 16 nm with an increase in iron content to 10%. Additionally, a progressive distortion in the nanoparticle shape, developing a more elongated form, was noted [54].

In our earlier investigation [48], we observed that nanoparticle sizes decreased from 62.8 nm to 39.9 nm upon the addition of iron to titania. Iron exerts a crystal growth suppression effect on TiO_2_, hindering particle contact and inhibiting crystal growth during heat treatment [6]. Given the slightly smaller radius of Fe^3+^ compared to the channels along the c-axis of pure TiO_2_ (0.77 nm), Fe^3+^ may preferentially diffuse along the c-axis [60], substituting for Ti^4+^ in the TiO_2_ lattice and causing lattice expansion due to elongation of the tetragonal cell parameter [22]. This leads to lattice deformation [38], impeding grain growth. An increase in iron content substantially alters the morphology of TiO_2_ nanoparticles, causing them to lose their characteristic spherical shape and appear more agglomerated, although low iron levels have a minimal impact on morphology [61].

N. Abbas et al. (2016) reported that in samples with high iron content (35%), fine granular nanostructures became larger and more dispersed, as shown in SEM images (Figure 4).

The crystallite size increased from 21.5 to 62.1 nm, and the shape changed to a mix of spherical nanoparticles and rod-shaped structures with a length of approximately 84 nm [56] as lighted with red lines.

In summary, the morphology of titanium dioxide nanoparticles is significantly influenced by iron content, impacting their dimensions. Specifically, the addition of iron reduces both crystallites’ [55] and nanoparticles’ [52] sizes. This alteration is often accompanied, although not universally, by an increase in specific surface area. Beyond a certain threshold (around 10%, corresponding to a Fe/Ti ratio of 0.1), the reverse process occurs, leading to nanoparticle enlargement, likely induced by the transformation of anatase to rutile activated by iron [62]. In any case, iron contributes to rendering the nanoparticles more heterogeneous, causing them to deviate from their characteristic spherical shape.

### 3.2. Band Gap of Semiconductor

The incorporation of heteroatoms into the crystal structure of semiconductors, such as titanium dioxide, has the potential to decrease their band gap [63]. This is particularly evident when iron is integrated into the lattice of titania. Absorption below 380 nm (approximately 3.1 eV) is associated with the intrinsic band gap absorption of pure TiO_2_ [64] (3.0 eV for the rutile phase and 3.2 eV for the anatase phase [41]). Iron doping induces absorption in the visible region, typically at 400 and 500 nm, intensifying with increasing iron content. These absorptions are linked to the excitation of 3d electrons of Fe^3+^ ions into the conduction band of TiO_2_ (charge transfer transition) [65], resulting in a band at around 400 nm [22]. Iron does not impact the valence band of TiO_2_ [45]. For higher iron concentrations, an additional band around 500 nm is observed, possibly due to the d–d transition of Fe^3+^ [38,66] or the charge transfer transition between interacting iron ions (2Fe^3+^ → Fe^4+^ + Fe^2+^) [22], with the maximum interaction occurring at approximately 1.5%.

Furthermore, iron induces the formation of structural defects, including oxygen vacancies [44]. Oxygen vacancy formation allows electrons to migrate from O 2p states in the valence band to Ti 3d states in the conduction band. Ti^3+^ states, produced by trapping electrons in defective sites, accumulate and reflect the number of defect sites [44]. Each Ti atom surrounding the removed oxygen can capture one of these electrons, forming Ti^3+^-related defect states within the bandgap, reducing the conduction band [44]. Ti^3+^ sites, increasing with decreasing nanoparticle size, serve as photocatalytically active sites and behave as coordinatively unsaturated ions. Therefore, their study, particularly in photooxidation and photocatalysis, deserves special attention [67].

The introduction of iron into the crystalline structure of titania significantly reduces the band gap (Figure 5). Even minor amounts of iron (up to a molar ratio of Fe/Ti of 0.1) drastically reduced the band gap from the 3.2 eV of pure titania to approximately 2.1−2.5 eV [6]. These values make the catalyst active under visible radiation [40], and the reduction in the band gap is due to the formation of the Fe–O–Ti bond in the crystal lattice of TiO_2_ [51].

Beyond a molar ratio of Fe/Ti of 0.1, a stabilization of the band gap is observed around approximately 1.9–2.1 eV, approaching the band gap value of hematite. This implies that limited quantities of iron (<10%) are required to reduce the band gap. Higher quantities do not induce a drastic decrease because iron, beyond this percentage, is no longer effectively incorporated into the crystal lattice of titania, surpassing the solubility limit of iron in titania. Excess iron tends to form a separate phase on the surface of titania, primarily as hematite (α-Fe_2_O_3_) [31,61], goethite (α-FeOOH) [54], ilmenite (FeTiO_3_) [35], or Fe_2_TiO_5_ (pseudobrookite) [29,34], especially favored at high calcination temperatures [38] (800–1000 °C [29]). The coexistence of multiple phases cannot be excluded, as observed by our previous work [48] with 2% iron loading. Notably, iron exhibits a higher tendency than other transition metals (Co, Cr, Cu, and Mo) to form separate phases as iron oxide [45]. Attention is required for the segregation of surface iron, as Fe_2_TiO_5_ and Fe_2_O_3_, despite possessing a reduced band gap (2.18 eV [29] and 2.00–2.20 eV [32], respectively), exhibit low photocatalytic activity and could occupy active sites dedicated to the adsorption of contaminants and photocatalytic degradation. Therefore, meticulous control of the dopant quantity and distribution is crucial.

Several authors have reported a solubility limit of around 1% (for anatase) using conventional doping techniques [22,36,43,63]. For instance, D. Cordischi et al. (1985) found a Fe solubility of about 1% in anatase using wet preparation techniques, such as co-precipitation followed by high-temperature calcination. The solubility in rutile was only 0.1% (Fe/Ti 0.001) [34]. Iron is less soluble in rutile than in anatase due to the latter’s symmetric nature, resulting in additional tunnels with larger volumes. These tunnels can accommodate cations or anions of significant dimensions introduced during crystal synthesis [79], contributing to greater solubility in anatase [29,37].

Li X. et al. (2003) observed a solubility limit of 1 at. % Fe in anatase prepared by a sol-gel process followed by annealing at 450 °C for 2 h [43]. Z. Zhang et al. (1998) reported that up to a 1% iron content in TiO_2_ using the sol-gel method followed by hydrothermal treatment did not result in secondary phases [42]. Litter M. (1996) noted that iron initially present on the surface diffuses into the bulk, producing a solid solution. Samples containing up to 1% Fe substitutional solid solutions with dispersed Fe^3+^ in the TiO_2_ lattice. Higher iron contents host excess iron in the form of tiny particles or small aggregates of iron oxides (hematite) and/or mixed oxides (Fe_2_TiO_5_) on the surface of the solid solution particles [29].

Even with less conventional techniques like the solid-state method, a separate iron phase was observed at 1% iron [48]. A more in-depth study revealed that approximately 10% of the introduced iron formed the segregated phase, with the remaining fraction effectively incorporated into the TiO_2_ lattice [80].

W. Teoh (2007) observed iron segregation at Fe/Ti ratios of 0.05, notably higher than in previously reported studies, due to the extreme conditions of doping through flame spray pyrolysis [27]. Z. Wang et al. (2001), using the same technique, demonstrated the formation of the Fe phase at a Fe/Ti ratio of 0.10 [81]. Flame spray pyrolysis (FSP) may yield a higher solubility limit of Fe in TiO_2_ compared to conventional techniques.

Solubility is also influenced by the type of TiO_2_ and its predisposition to incorporate iron. Using Degussa P25, the iron-segregated phase formed at Fe/Ti ratios as low as 0.005, consistent with the observed band gap [48]. The reduction in band gap was less pronounced, indicating that Degussa P25 did not incorporate Fe ions into its lattice, resulting in fewer bonds responsible for reducing the Fe–O–Ti band gap.

This phenomenon could be due to the homemade titania having more structural defects compared to Degussa P25, facilitating greater ease for Fe^3+^ ions to diffuse into the crystalline lattice. This resulted in a more significant decrease in the band gap [60]. Another critical factor influencing diffusion is the calcination temperature during synthesis. Elevated temperatures (above 300 °C, with a recommended range of 600–700 °C) promote the diffusion of iron into the crystalline lattice of titania [60]. This is crucial for preventing non-uniform diffusion, primarily near the catalyst’s surface, and facilitating diffusion towards the center.

In addition to calcination temperature, the doping strategy also influences diffusion. For instance, Sonia J. (1991) demonstrated that samples prepared via co-precipitation allowed for less iron diffusion into the lattice compared to the impregnation method [36].

In summary, the solubility limit of iron in the TiO_2_ structure is influenced by doping strategy, material characteristics (such as the presence of structural defects), and synthesis methodology. Generally, the solubility limit is around the Fe/Ti ratio of 0.01. However, a significant reduction in band gap occurs up to Fe/Ti ratios of 0.1. This suggests that the segregated iron on the photocatalyst surface also contributes to the band gap reduction [35], likely due to the formation of Fe–O–Ti bonds at the heterojunction of the two distinct phases. However, this does not consistently translate to improved photocatalytic performance.

### 3.3. Surface Area

The presence of low iron concentration inhibits the growth of the titanium dioxide crystalline lattice [82,83], resulting in smaller grain size (pore) and increased surface area, which is beneficial for photocatalysis by enhancing contaminant adsorption, especially in mesoporous structures, providing more active sites for photocatalytic reactions [49]. For instance, J. Zhu et al. (2006) observed a rise in the specific surface area of TiO_2_ nanoparticles with 0.5% iron from 140 to 159 m^2^/g [38]. Similarly, C. Almquist (2002) reported an increase in surface area at an iron content of 0.5% from 69 m^2^/g for the pure TiO_2_ sample to 120 m^2^/g [49].

N. Jamalludin (2011) observed an increase in titania surface area with low iron content; pure TiO_2_ exhibited 43 m^2^/g, while 0.4 mol % Fe^3+^ incorporation increased the surface area to 74 m^2^/g [84], reaching 85 m^2^/g at 0.8 mol % Fe^3+^. Beyond 1.0 mol% Fe^3+^, the surface area decreased from 85 to 73 m^2^/g.

Careful consideration of iron concentration is crucial, as excessive iron, beyond the solubility limit, may deposit as oxide [34], causing pore blockage [74] and decreasing surface area [69,85].

Shi X. (2019) observed a decrease in surface area from 120 to 107 m^2^/g with an increase in iron concentration from 0.5 to 2.0%. Additionally, the pore volume reduced from 0.26 to 0.21 cm^3^/g, and the average diameter decreased from 10.8 to 10.5 nm [74].

I. Ganesh et al. (2012) experienced an increase in surface area from 23.25 m^2^/g to 65.92 m^2^/g with 0.1% weight of Fe [86], but a gradual decrease with increasing concentration, reaching 48.82 m^2^/g at 10% due to the pore blockage of the support by metallic oxides [45]. Ao J. (1999) observed a progressive decrease in surface area from 57 to 45 m^2^/g with 0.5 to 5.0% iron loading [37].

Z. Li et al. (2008) observed a smoother surface with Fe-doped TiO_2_ nanoparticles, leading to decreased specific surface area [52] from 113 m^2^/g for undoped TiO_2_ to 111 and 108 m^2^/g, respectively, for 1% and 5% Fe. Y. Zhang et al. (2021) reported reduced surface areas and porous volumes with >10% iron loading due to denser α-Fe_2_O_3_ nanoparticles occupying the interparticle voids of TiO_2_ [61].

Adan C. (2006) stated that beyond 0.7%, excess iron segregates, occluding the pores and reducing their volume, average diameter, and surface area [22].

However, the surface area appears to be weakly dependent on iron, as shown in Figure 6.

Iron has minimal impact on surface area beyond a threshold Fe/Ti ratio of 0.1, with only a slight decrease. Calcination temperature significantly affects surface area, necessitating control below 600 °C and iron concentration below 0.5% for optimal photocatalysts [93]. M. Qamar et al. (2014) found the highest photocatalytic activity at 450 °C, indicating a compromise between surface area and crystallinity [94]. M. Litter (1996) reported a similar result: decreasing surface area with increasing calcination temperature, emphasizing the need to control dopant concentration and optimize temperature between 450 and 600 °C for a large surface area (Figure 7).

The surface area decreased as the calcination temperature increased. Specifically, for the impregnation method, the surface area decreased from approximately 30 m^2^/g (at 500 °C) to 1 m^2^/g (at 1000 °C). A similar trend was observed for samples prepared using the co-precipitation method, albeit with less consistency in surface area due to different preparation methods. In contrast, the iron content, while introducing variations, did not significantly influence the surface area, except for iron contents between 0.5 and 2%. For instance, using the impregnation method, the surface area of the sample calcined at 650 °C decreased from 17 m^2^/g (with 0.5% iron content) to 9 m^2^/g (for the sample containing 5% iron). It is advisable to carefully control the dopant concentration, as an excess, likely above 0.5%, causes a moderate decrease in specific surface area due to pore occlusion. Furthermore, the optimal temperature for achieving a large surface area appears to be between 450 and 600 °C.

### 3.4. Anatase to Rutile Transition

In addition to the calcination temperature, which promotes the formation of anatase beyond 370 °C [43] and the formation of rutile beyond 500 °C [79], considering the treatment time is crucial since the process is time-dependent, involving a reconstructive-type transformation [95,96]. Furthermore, iron is recognized as a promoter of rutile formation [97] more than other elements such as V and Cr [35]. This phenomenon is particularly evident when these elements are dissolved in a solid solution [29]. The rutile phase can be observed even at 400 °C during calcination when iron is present in high concentrations (>2% [43]).

The anatase phase is preferred for photocatalytic [7] and photovoltaic reactions due to its crystalline structure [41]. The higher band gap promotes the swift migration of charge carriers and inhibits positive hole–electron recombination [96]. This is owing to the specific structure with slightly greater distortion in the anatase phase compared to the rutile phase. In rutile, the energy level of the conduction band closely aligns with the reduction potential of O_2_, delaying oxygen radicalization [10]. Additionally, anatase possesses a slightly higher Fermi level and a higher degree of surface hydroxylation compared to rutile, and its formation occurs at a lower temperature (T < 600 °C). This results in a larger available surface area for absorption and catalysis [98,99,100]. Furthermore, anatase is widely available and cost–effective. Therefore, iron could be perceived as an antagonist to photocatalysis.

Trivalent iron (Fe^3+^) induces oxygen vacancies to maintain local charge neutrality, given its lower oxidation state compared to Ti^4+^ in the titanium dioxide structure, promoting rutile formation, as commonly observed with ions having a smaller radius and lower valency than Ti^4+^ [46]. The presence of oxygen vacancies makes the material less “rigid” and more prone to reorganization of bonds towards thermodynamically more stable states, such as rutile [96].

Iron induces relaxation of the apical Ti−O bond in anatase, inhibiting the formation of surface Ti^3+^ due to its multi-valency. Transitioning between Fe^2+^ and Fe^4+^ can balance charge excesses, promoting the phase transition from anatase to rutile. The ability of iron to hinder the formation of surface Ti^3+^ and induce Ti−O bond relaxation facilitates this transition. Elevated surface Ti^3+^ levels hinder the phase transition from anatase to rutile [28].

Furthermore, the multivalence of iron induces additional oxygen vacancies as it has the capability, during high-temperature calcination, to undergo reduction from Fe^3+^ to Fe^2+^, causing the oxidation of oxygen to molecular oxygen. Experiments with Al^3+^ [28], having the same valency as Fe^3+^ and a similar ionic radius, lead to the inhibition of rutile formation. The use of Fe^2+^ also fails to promote rutile formation as it cannot induce oxygen vacancies in the titanium dioxide lattice [96].

Nasralla N. (2020) observed that, at the same calcination temperature (600 °C), the anatase phase was 17% with a 3% iron load but completely disappeared (quantitatively converting to rutile) at 8% iron load [41].

The significant role of iron and, notably, the decisive influence of the calcination temperature is evident in Figure 8. Notably, the specific duration of calcination, typically ranging from 1 to 4 h, has not been considered, except in the study by J. Navio (1998), where the sample was subjected to a 24-h treatment at 500 °C [37].

For excessive amounts of iron, the rutile phase decreases as non-negligible pseudobrookite formation occurs [103] due to iron oxide segregation.

However, a mixture of anatase and rutile phases is optimal for photocatalytic processes compared to single-phase crystals [105,106]. The enhanced activity is due to the heterojunction formation between different phases, efficiently separating spatial charges and, consequently, improving the TiO_2_ quantum yield [107]. Given the slightly lower conduction band of rutile than anatase (approximately 0.2 eV), photogenerated electrons preferentially transfer from anatase to the rutile band, while positive holes follow the reverse process [21], leading to distinct photogenerated charges separation. Therefore, the mixture of anatase and rutile phases can effectively suppress the electrons and holes recombination, localizing them in rutile and anatase, respectively [41].

The intimate contact between the two phases is crucial to enhance the photocatalytic activity of mixed-phase TiO_2_ [108]. Numerous studies have demonstrated that there exists an optimum ratio between the two phases of TiO_2_ to maximize photocatalytic activity, highlighting the need to explore effective methods for synthesizing mixed-phase TiO_2_ with close contact and a suitable ratio of the two phases [106].

Materials with a higher anatase/rutile ratio, typically 80:20, such as Degussa P25, a nanocrystalline material obtained through flame pyrolysis [96], exhibit superior photocatalytic activity [13,61]. A partial conversion towards rutile is desirable, even if induced by iron, as observed in a study by R. Bacsa (1998) where multiphase material of 70% anatase and 30% rutile showed enhanced performance [109].

High quantities of iron disrupt the octahedral structure of TiO_2_ to such an extent that it leads to the formation of an amorphous phase in the overall material structure [35,81].

### 3.5. Charge Recombination

The trivalent iron (Fe^3+^) acts as a trap for both positive holes and photogenerated electrons, thereby increasing the lifetime of the photogenerated pair [52] and inhibiting their recombination [42]. Typically, the lifetime of photogenerated charges in titanium dioxide is around 200 microseconds [110], with the transfer of the photogenerated electron from TiO_2_ to O_2_ ranging from microseconds to milliseconds [44]. However, iron addition (0.5%) increases the lifetime to 50 milliseconds [63]. Higher charge carrier concentration induces a corresponding increase in photocatalytic reactivity [64], posing a crucial aspect in overcoming bottlenecks in photocatalytic processes [44]. Additionally, iron induces oxygen deficiencies [41] and undercoordinated titanium atoms in the TiO_2_ structure, acting as an electron trap [28]. However, if an electron is trapped in a deep trapping site, it may have a longer lifetime but could also have a lower redox potential, potentially decreasing photocatalytic reactivity. The reactivity of doped TiO_2_ appears to be complex, dependent on dopant concentration, the energy level of dopants within the TiO_2_ lattice, their d-electronic configuration, the distribution of dopants, and the concentration of electron donors [110]. This effect is based on the presence of distinct energy levels resulting from the different oxidation states of iron (Fe^4+^/Fe^3+^) located above the valence band of TiO_2_ and below the conduction band of pure TiO_2_ [111].

During the initial step of the photocatalytic process, Fe^2+^ ions are produced as excess photogenerated electrons mitigate from pure TiO_2_ to Fe^3+^ [36]. Due to its instability arising from electron configuration loss (d5), Fe^2+^ swiftly oxidizes to Fe^3+^, transferring electrons to the adsorbed O_2_ and forming superoxide ions [38,43]. Iron is an excellent electron acceptor compared to Ti^4+^, even for electrons released from the reverse process [36]. At the same time, Fe^3+^ can act as a trap for positive holes, considering its energy level (Fe^3+^/Fe^4+^) located above the valence band of TiO_2_. Consequently, Fe^4+^ ions accept electrons and revert to Fe^3+^, while −OH groups transform into hydroxyl radicals. Bulk Fe^3+^ ions facilitate efficient charge separation by temporary electron trapping and extending the lifetime of positive holes, through surface−OH groups, producing hydroxyl radicals [36]. Charge carriers trapping (h^+^, e^−^) in different Fe^3+^ sites significantly enhance charge separation and interfacial charge transfer, thus enhancing photocatalytic efficiency [43].

However, beyond the optimal concentration, reported around 0.5% by W. Choi et al. (1994) [63] and 1% by Li X. (2003) [43], results in an exponential increase in the recombination rate. This is attributed to a decrease in the average distance between trapping sites due to the increasing number of dopants within the nanoparticle. The increased metal–metal interactions and energy transfer between nearby ions lead to the dominance of the concentration quenching process, becoming the predominant non-radiative decay process [41,47], reducing photocatalytic reactivity [38].

High iron loading indices additional phases, consequently, form a heterojunction between different phases [29]. This negatively impacts photocatalytic activity by increasing the recombination rate [32] and reducing charge mobility. Positive holes diffusing only 20–40 Å due to rapid recombination are particularly affected. Furthermore, in hematite, positive charges could be trapped in deep states such as Fe^4+^, significantly reducing their oxidizing power.

However, the heterojunction facilitates bidirectional charge transfer, leading to a distinct separation between positive holes and photogenerated electrons, reducing the charge recombination probability [18,61]. This occurs because the oxidation and reduction reactions take place separately in two different areas of the photocatalyst [8,21]. Specifically, electrons generated in the hematite conduction band migrate toward the TiO_2_ conduction band, reacting with the adsorbed oxygen on the catalyst surface and producing the superoxide radical. Simultaneously, positive holes photogenerated in the TiO_2_ valence band migrate towards the Fe_2_O_3_ valence band, reacting with adsorbed water to generate the hydroxyl radical [112]. Therefore, the formation of a segregated iron oxide phase is not necessarily a negative phenomenon. It essentially represents a coupling between TiO_2_ and iron oxide, a strategy widely used to enhance the photocatalytic performance of semiconductors by coupling them with semiconductors with a lower band gap, such as Fe_2_O_3,_ in this case (2.1 eV) [113]. An example is the study carried out previously [80], where selective leaching was performed on a titanium-based photocatalyst doped with iron to remove the segregated iron on the surface. The results showed a significant decrease in photocatalytic performance in the removal of rhodamine B.

### 3.6. pH_ZC_

The pH of the environment significantly influences the overall efficiency of photocatalytic processes [98]. pH dependence can be associated with changes in the surface charge of the photocatalyst, hydrophobicity, net charge of pollutants, changes in adsorption modes, and the amount of produced hydroxyl radicals, thereby modifying the overall rate. Additionally, variations in pH may introduce deactivation problems if the presence of long-lived intermediates that inhibit the photocatalyst is favored [114]. The interaction of electron donors and acceptors with metal oxide semiconductors is influenced by surface chemistry and by the point of zero charge (pH_ZC_), representing the pH value where the coverage of H^+^ equals the coverage of OH^−^. The pH_ZC_ is closely related to the surface acidity of a solid, and its knowledge allows for the evaluation of a surface propensity to become either positively or negatively charged as a function of pH, making it an essential parameter for defining or modulating the activity of a photocatalyst. Regarding TiO_2_, the principal amphoteric surface functionality is the “titanol” moiety, –TiOH. Hydroxyl groups on the TiO_2_ surface are known to participate in an acid−base equilibrium, where pKa1 and pKa2, constants of the first and second acid dissociation, at the pH_ZC_ for Degussa P25, the corresponding surface acidity constants were found to be 4.5 and 8.0 respectively, resulting in a pH_ZC_ of 6.25 [110,115]. This implies that interactions with cationic electron donors and electron acceptors will be favored at pH > pH_ZC_ conditions, while anionic electron donors and acceptors will be favored at pH < pH_ZC_ conditions [110]. Adsorption of relatively nonpolar pollutants, such as 1,2-diethyl phthalate, is enhanced at pH close to pH_ZC_. On the other hand, the difference in pH_ZC_ values among various TiO_2_ photocatalysts could affect the reaction mechanism. Furthermore, both the type and the amount of dopant metals influence this value [91]; the pH_ZC_ values of iron-containing titania increase significantly with the increase in metal content, indicating a surface enrichment of species with basic behavior such as iron oxides (hematite) and goethite. Iron modifies the photocatalyst pH_ZC_, making it more basic [45]. For instance, A. Di Paola (2002) reported that the pH_ZC_ of titanium dioxide increased from 7.1 to 7.4 with 1% iron and further to 8.1 with 5% iron [91]. This implies that iron, by modifying the pH_ZC_ and subsequently altering the surface charge of the catalyst, can change the interaction between the catalyst surface and the target molecule. It can either promote or hinder the adsorption of the contaminant, thereby influencing the photocatalytic performance, significantly influenced by the adsorption step necessary for the target molecule to undergo photodegradation [36,116]. Hydroxyl radicals, due to their instability, have a short half-life, allowing them to diffuse only about 180 Å into the liquid bulk [117]. Therefore, they react only with molecules near the catalyst surface. The affinity of the target species towards the photocatalyst is crucial for effective photocatalytic degradation. This aligns with A. Di Paola (2004) observed that an increase in the amount of iron led to an increase in pH_ZC_ and, consequently, an increase in the interaction and degradation rate of formic acid. A similar effect for benzoic acid was observed, although steric hindrance led to interactions not solely dependent on pH_ZC_. Due to acetic acid’s low acidity, a linear correlation between the surface basicity of the catalyst and the degradation of the target molecule was not observed [45].

### 3.7. Photocatalytic Performance

The photocatalytic performance is heavily influenced by the previously discussed parameters, which, in turn, are affected by the iron content. Establishing the dependence of photocatalysis on these parameters is very complex due to their interdependence and the various phenomena involved. Nevertheless, various authors have studied photocatalytic performance concerning iron content in the photocatalyst using widely varying reaction conditions and target molecules for degradation. For example, different irradiation employed, a parameter with a substantial influence on photocatalysis, complicates direct comparisons across studies [118]. Photocatalysis is also influenced by contaminant type and concentration, catalyst dosage, and solution turbidity. The solution pH can also impact the yield of the process; M. Nazari (2019) observed that at pH 4.5, the best-performing catalyst had 0.7 mol% Fe, while at pH 2.4, it was 3 mol% Fe [119].

Several authors agree on the optimal iron dosage that yields the best performance. For instance, Moradi H. (2016) reported that the optimal Fe^3+^ dosage for Reactive Red 198 degradation, in an investigation range between 0 and 10%, was 1%, attributing the performance decrease beyond this dosage to excessive recombination of photogenerated charges [6]. Ghorbanpour M. (2019) reported that the optimal performance in methyl orange degradation was achieved using a catalyst containing 0.5% Fe synthesized via the molten salt method [120]. Ochoa Rodriguez P. (2019) observed that for acid orange 7 degradation, the optimum was reached with 0.1% Fe [76]. The authors attributed these optimal performances to relatively low Fe/Ti ratios, ensuring that iron was finely dispersed in the crystal lattice of TiO_2_.

In the phenol degradation, Adan C. (2006) observed that the best performance was achieved with an iron dosage in titania of 0.7–1%, corresponding to the maximum concentration of iron incorporable into the crystal lattice of titania without forming segregated phases of iron oxide [22]. Similarly, in the oxidation of cyclohexane, Li X. (2003) agrees on an optimal iron dosage of 1%, attributing the decrease in performance beyond this threshold to the formation of segregated iron oxide phases on the surface that deactivate active sites dedicated to photocatalytic degradation [43]. The deleterious effect of surface iron on photocatalytic activity was convincingly demonstrated by V. Moradi et al. (2019). Their research showcased that the removal of surface iron through selective leaching significantly enhanced the catalyst’s efficiency in degrading phenol [121].

Furthermore, our previous study [80] has demonstrated, in contrast with V. Moradi, that surface−segregated iron was beneficial for enhancing photocatalytic performance. Specifically, removing the surface iron corresponding to approximately 10% of the total iron resulted in a decrease in the removal kinetics of rhodamine B, likely due to a reduced ability of the material to effectively separate photogenerated charges and prevent the recombination process, which is probably the rate-determining step under the employed conditions. This observation implies that the optimal amount of iron is contingent upon the specific contaminant. Consequently, the atom% of iron in TiO_2_ corresponding to the maximum photocatalytic performance is documented in Table 1 through a comparative analysis of various studies.

Table 1 reveals a notable discrepancy among authors regarding the optimal iron dosage for methylene blue, ranging from 0.1 to 4.7%. In contrast, there is greater consensus for phenol and methyl orange, with optimal ranges of 0.1–0.6% and 0.1–0.5%, respectively. This observation holds interesting implications; varying iron dosages could potentially enable the synthesis of a selective photocatalyst tailored to a specific contaminant.

Soria J. (1991) reported that in the reduction of molecular nitrogen for ammonia production, the most performing materials were found to have iron content of 0.2% and 0.5% [36]. In these materials, no surface–segregated iron was present. Therefore, the observed photocatalytic activity was mainly due to Fe^3+^ ions diffused in the crystal lattice of TiO_2_. The decrease in activity with increasing surface enrichment in Fe suggests that surface iron negatively influences specific phases of the mechanism and/or specifically blocks certain surface sites. Since only non-surface Fe is active (Fe_2_O_3_ and Fe_2_TiO_5_), its role must be linked to the photogeneration and/or evolution of electrons and holes in TiO_2_ grains. The observed influence of these iron ions in the photoreduction of dinitrogen can be explained by the hypothesis that they trap electrons, promoting their separation from the holes and allowing the latter to reach the surface. This, in turn, reacts with hydroxyl groups, giving rise to highly reactive hydroxyl radicals through surface chemical reactions. It is interesting to compare this result with the findings of Cordischi D. (1985), as they reached the same conclusions [34]. However, the most efficient material for ammonia production was found to have iron content of 0.5% and 1%. In this case, too, a decrease in photocatalytic performance was due to surface-segregated iron (Fe_2_TiO_5_). This concludes that surface-segregated iron negatively affects the process. However, the tendency to segregate depends on different synthesis conditions. Soria J. and D. Cordischi obtained different solubility limits, 0.5% for the former and 1% for the latter. However, Litter M. (1996) reported that the best performances related to the reduction of molecular nitrogen are achieved at iron dosages of 0.2% [29].

The consideration that the band gap decreases monotonically with increasing iron content up to a Fe/Ti molar ratio of 0.1 is noteworthy. This implies that lower band gaps do not necessarily correspond to better photocatalytic performance, suggesting the existence of other relevant phenomena.

To summarize, most authors analyzed in this review agree that surface-segregated iron oxide is detrimental to photocatalytic performance. Therefore, the optimal value of Fe to be employed often coincides with the maximum solubility limit in titanium oxide. However, it is important to emphasize that solubility, and thus the ability of titanium oxide to incorporate iron into the crystalline bulk, is closely dependent on the synthesis method employed, the calcination temperature, and the presence of anatase and rutile phases.

## 4. A Possible Optimized Synthesis of Nanoparticles Based on Iron-Doped TiO_2_

Upon reviewing the scientific literature, a potentially optimized synthesis strategy for producing iron-doped titanium-based nanoparticles for photocatalytic treatments under visible light has been suggested. It was observed that the most suitable nanoparticle size ranged from 25 to 30 nm [49], achievable with an iron content of approximately 0.5%. This parameter appears independent of the synthesis strategy, although various methods were explored, with the majority employing the sol–gel method [50,55,127], followed by co-precipitation [128], hydrothermal [52], and solid-state methods [48].

The chosen synthesis strategy was found to potentially impact the solubility of iron, a critical factor in maximizing photocatalytic performance. Most authors identified the optimum point when the iron content in the titanium crystal lattice reached its maximum without forming segregated phases of iron oxides on the catalyst surface, which would compromise its activity [22,43]. Optimal performance was generally reported with iron content ranging from 0.5% to 1% [6,34,129], corresponding to the solubility of iron in anatase [42]. Exceptions, such as the study by Z. Wang (2001) and W. Teoh (2007) utilizing the spray pyrolysis strategy, demonstrated the possibility of achieving catalysts with 5–10% iron content without segregated phases [81], although performance was in line with other photocatalysis works (kinetic constant, K = 0.36 h^−1^) [27]. However, a precise comparison is challenging, given that reaction kinetics depend heavily on surrounding conditions, such as the light source used, the type of contaminant, and its concentration.

The impact of iron on surface area is minimal, with calcination temperature exerting a more pronounced influence [29]. Elevated calcination temperature leads to a decreased specific surface area. It is advisable to operate at relatively low temperatures, preferably above 370 °C, to facilitate anatase formation (the active phase of titanium) and below 500 °C to avoid excessive annealing processes and rutile formation. A desirable outcome involves a mixture of the two phases (20–30% rutile), proving more active than pure anatase [13,61]. Given that iron promotes the anatase–rutile transition and considering the time-dependent nature of the process, a recommended calcination temperature is approximately 450 °C [94] for 1–3 h.

Up to an iron content of 0.5–1.0%, the expected band gap is around 2.8 eV for 0.5% Fe and 2.7 eV for 1% Fe, suitable values for operation under visible light radiation. The precursor of iron and titanium does not seem to have relevance (for TiO_2_, homemade production is preferable, as it is more easily dopable than Degussa P25 [48]).

Given the considerations and the fact that the synthesis of titania nanoparticles and doping with iron need not occur in two steps, a representative schematic of the proposed process has been illustrated in Figure 9.

For future commercial and research applications, especially in the water treatment field, it will be necessary not only to synthesize a high–performance photocatalyst in a simple and environmentally friendly manner able to operate under visible light but also to develop an effective strategy to iin my opinion they do not need an explanation in the text, they serve to indicate the order of the stepsmmobilize the photocatalytic nanoparticles onto the surface of the supporting material [130,131] (to avoid costly phases of catalyst separation from the treated water [132,133]). It is worthwhile to investigate how efficacy varies with operating conditions, aiming to identify the optimal catalyst, whose properties can be modulated based on the iron amount (or different dopants) used, depending on the effluent, conditions, and the target molecule, resulting in selective degradation. Additionally, it is essential to define accurate criteria for the selection of a photocatalytic reactor [134] with the optimal geometry necessary to provide effective irradiation [11].

## 5. Conclusions

The impact of iron on various parameters governing photocatalysis has been analyzed. It was observed that the presence of iron tends to reduce the nanoparticle size. However, beyond a threshold value, the reverse process occurs, as the initially spherical nanoparticles tend to elongate due to the rutile phase induced by iron. Indeed, iron serves as an excellent promoter of the anatase–to–rutile transition, detrimental for the material since the anatase phase is more photocatalytically active. However, a mix of the two phases is recommended to achieve superior performance.

Iron diminishes the band gap of the semiconductor from 3.1 eV to values comparable to hematite (1.9–2.1 eV), rendering the photocatalyst active even under visible light radiation. Specifically, up to Fe/Ti molar ratios of 0.1, the band gap consistently decreases with increasing iron content, settling around values of about 2.0 eV after surpassing this ratio of 0.1. Iron has the capability of extending the lifetime of the positively charged hole–electron pair, reducing the charge recombination rate, and consequently enhancing the material’s ability to produce hydroxyl radicals according to the photocatalytic mechanism.

Iron can alter the surface properties of titania by modifying the pH at the point of zero charge (pH_ZC_), thereby influencing interactions between the photocatalyst surface and contaminants. Furthermore, it slightly alters the surface area at low dosages, while at high dosages, it tends to decrease due to the formation of surface–segregated iron oxide that occludes the material’s pores. Therefore, the formation of such segregated phases should be avoided, following the examined photocatalytic performances, which were found to be optimal around the maximum solubility limit of iron in the titania lattice, typically around 1%, depending on the synthesis strategy and conditions, especially calcination temperature.

In conclusion, a potential synthesis/doping strategy to optimize all the studied parameters has been proposed.

## Figures and Tables

**Figure 1 nanomaterials-14-00293-f001:**
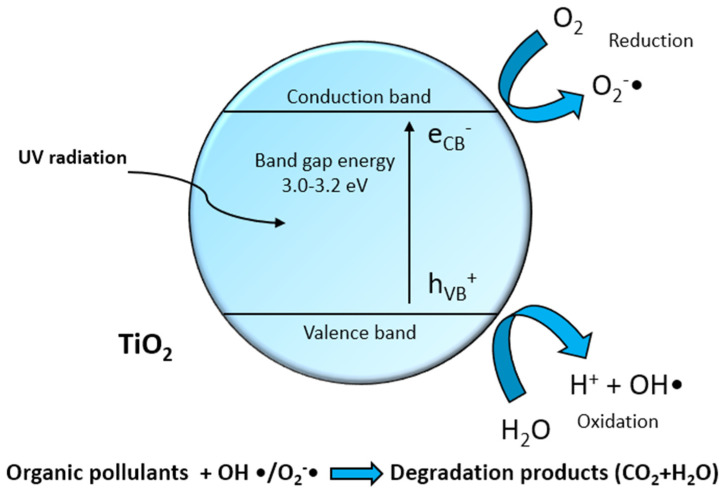
Schematic representation of the photochemical activation of titanium dioxide and the formation of radical species responsible for the oxidative degradation of organic pollutants.

**Figure 2 nanomaterials-14-00293-f002:**
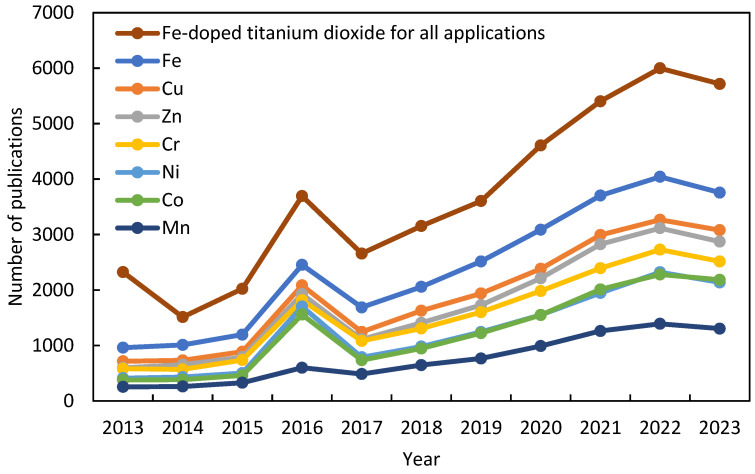
Number of publications on doped titanium dioxide with different transition metals for photocatalytic applications.

**Figure 3 nanomaterials-14-00293-f003:**
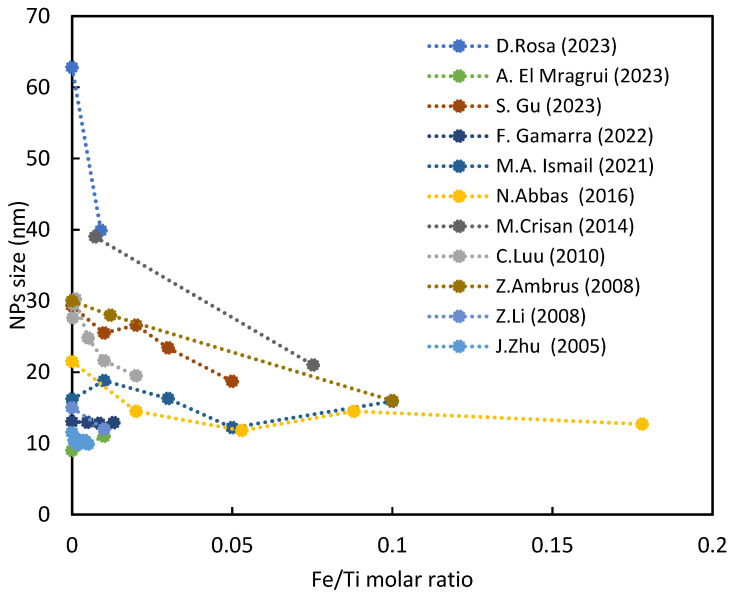
Size of TiO_2_ nanoparticles as the amount of iron used for different syntheses [38,48,51,52,53,54,55,56,57,58,59].

**Figure 4 nanomaterials-14-00293-f004:**
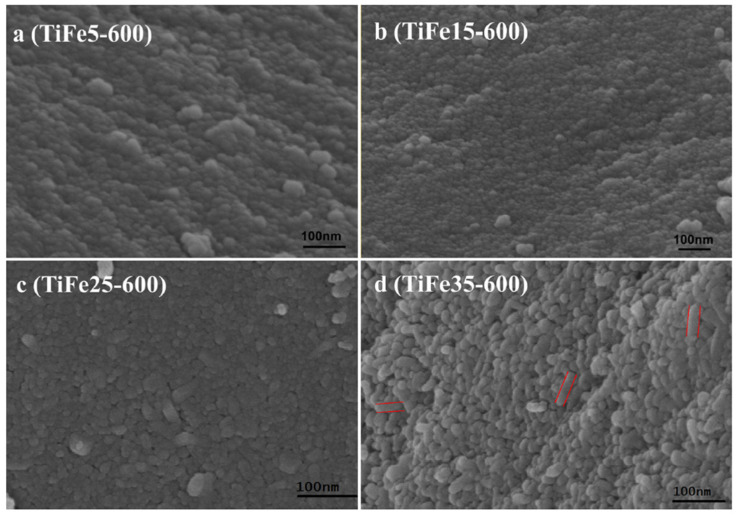
SEM micrography of based-TiO_2_ samples doped with different amounts of iron: (**a**) 5%, (**b**) 15%, (**c**) 25%, (**d**) 35% [56].

**Figure 5 nanomaterials-14-00293-f005:**
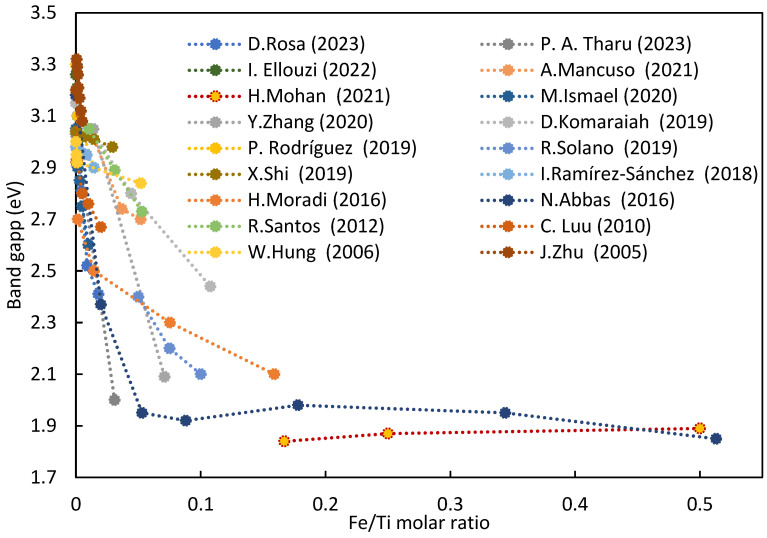
Band gap of titanium oxide doped with iron at varying Fe/Ti ratio [6,31,38,48,51,56,61,68,69,70,71,72,73,74,75,76,77,78].

**Figure 6 nanomaterials-14-00293-f006:**
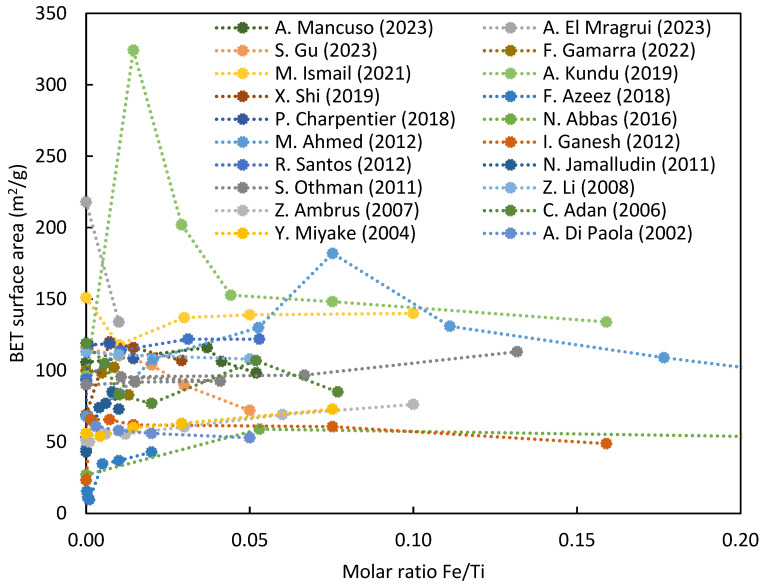
BET surface area of iron-doped TiO_2_ nanoparticles at varying Fe/Ti ratio [22,31,51,52,53,54,56,57,58,59,62,71,74,84,86,87,88,89,90,91,92].

**Figure 7 nanomaterials-14-00293-f007:**
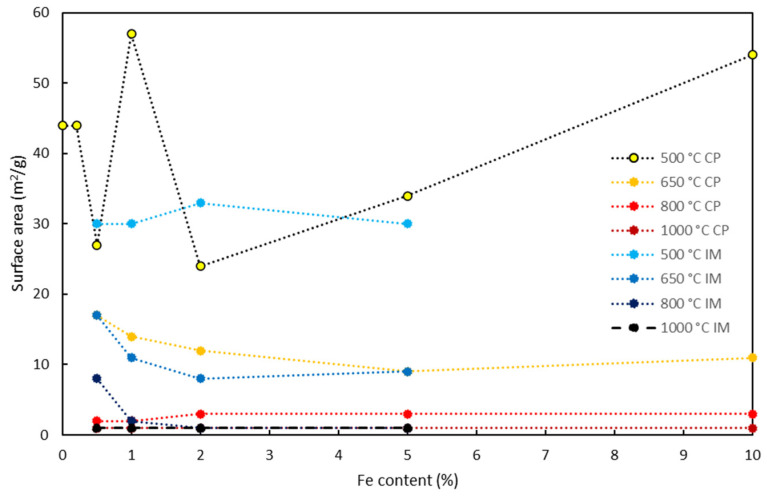
Surface area of Fe-doped titania samples prepared by impregnation (IM) and co-precipitation (CP) methods at different calcination temperatures [29].

**Figure 8 nanomaterials-14-00293-f008:**
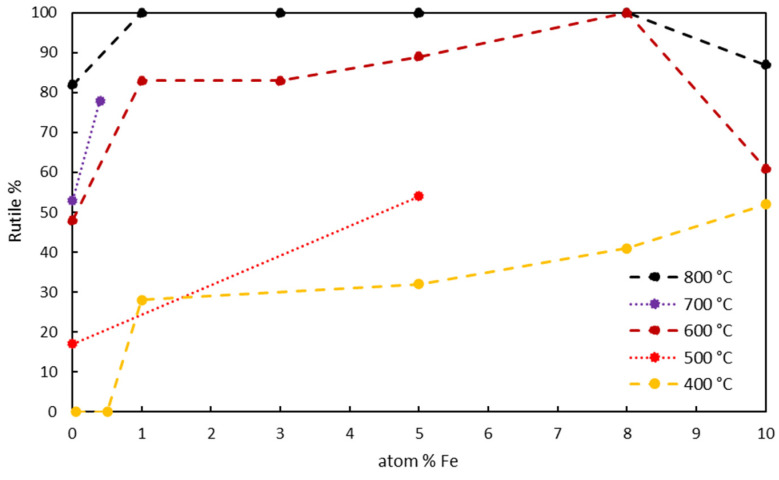
% of rutile phase in TiO_2_ as a function of iron content and calcination temperature [37,41,43,101,102,103,104].

**Figure 9 nanomaterials-14-00293-f009:**
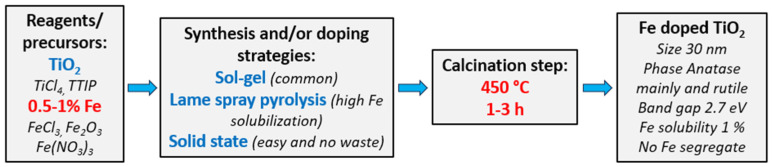
A possible optimized synthesis strategy to produce iron-doped titania nanoparticles for photocatalytic purposes.

**Table 1 nanomaterials-14-00293-t001:** Optimal atom% of Fe in TiO_2_, leading to maximum degradation, varied with different contaminants. This analysis considered only systems exposed to visible radiation and where the optimal % was verified within the range of % iron explored, excluding extremes.

Contaminant	Atom% Fe	Reference
Methylene Blue	0.1	[65]
	0.1	[122]
	0.3	[118]
	0.7	[119]
	0.9	[48]
	2.7	[70]
	3.0	[123]
	4.7	[56]
Acid Orange 7	0.1	[76]
	3.7	[71]
Methyl Orange	0.1	[73]
	0.3	[124]
	0.5	[120]
Phenol	0.1	[125]
	0.5	[121]
	0.6	[22]
Cyclohexane	1.0	[43]
Direct Blue 119	4.2	[126]

## Data Availability

The data presented in this study are available on request from the corresponding author.

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
