# Peer review of "Titanium Dioxide Nanoparticles Doped with Iron for Water Treatment via Photocatalysis: A Review"

_nanomaterials, 2024, doi:10.3390/nano14030293_

Round 1
Reviewer 1 Report
Comments and Suggestions for Authors
The manuscript is a review of iron-doped titanium dioxide nanoparticles for water treatment. The review is well-written and provides a clear picture of the major concern of the iron-doped titanium dioxide nanoparticles in the current study field. Overall, the manuscript is ready to be published in nanomaterials after several minor revisions.
1. In Figure 1, there are two Fe-related lines. The purpose of the second Fe plot is not quite clear. Please address.
2. In section 2.7, photocatalytic performance, due to the nature of the field, researchers are using various organic compounds to test TiO2, such as rhodamine B, acid orange, etc. Thus, the effectiveness of photocatalysis with various synthesis methods or conditions is not comparable. Could the authors narrow down the comparison by defining the performance?
3. The article lacks the SEM or TEM images from the cited publications, especially 2.1, making the reading of the description of the shapes with/without iron doping hard to flow.
Reviewer 2 Report
Comments and Suggestions for Authors
The authors comprehensively review the recent development of iron-doped titanium dioxide nanoparticles for photocatalytic applications, especially under visible light, are widely employed due to their promising performance. However, the manufacturing process, particularly the role of Fe3+ ions in the crystal lattice of titanium dioxide in operational parameters, may be controversial. Thus, this review aims to elucidate the role of iron, the optimal quantity, and how it influences key parameters of photocatalysis, including nanoparticle size, band gap, surface area, anatase-rutile transition, and point of zero charge. Furthermore, an optimized synthesis method is proposed based on data and considerations from past literature, focusing solely on iron-doped titanium oxide and excluding other types of dopants. This topic is very important yet overlooked by much of the research community. This Review Article is appreciated; however, the reviewer feels modifications are needed to further improve the manuscript, and this review can be considered for publication with following the next recommendation.
(1). The authors present a comparatively inclusive summary of the development, whereas the comments from the authors on these accomplishments are less elaborated. I suggest adding a summary of the benefits and drawbacks of each system/strategy.
(2). The authors should revise the introduction section incorporating the issues/developments of recent studies. The schematic illustration of photocatalyzer device/system should be added. Besides, the basic chemistry of and standards of photocatalysts and the comparative mechanism to their counterparts should be added in the revised manuscript.
(3). The author should add the section with heading “rise of advanced photocatalysts” with explaining the emergence and definite merits of this system than counterparts in the market.
(4). To arouse a broad interest from readership in this field, some recent literature on advanced photo/electrocatalysts systems should be included i.e., Materials Today, 2023, 67, pp. 203-228; ACS Catalysis, 2023, 13, 4, 2313-2325.
(5). It is suggested to add the prospects and/or comments on the commercial applications of this catalyst system. What are the challenges and promises of this system impending to the commercial applications?
(6). There are a lot of redundant descriptions which should be removed.
(7). The authors should improve the English language of the article, especially the sentence structures.
Comments on the Quality of English LanguageMinor changes/polishing is needed.
Reviewer 3 Report
Comments and Suggestions for Authors
1 - The English writing should be greatly improved. There are some typographical and grammatical errors in the manuscript. Hence, the manuscript should be checked carefully, and necessary corrections should be made.
2 - For a Review manuscript, the authors have very few references; Furthermore very recent works since 2022 on this topic have not been reported here.
I suggest you report more recent literature about the Fe doped TiO2, to improve the manuscript to be possible to be accepted for publication in the journal Nanomaterials.
Comments on the Quality of English LanguageThe English writing should be improved.
